# Stopping molecular rotation using coherent ultra-low-energy magnetic manipulations

Helen Chadwick [1] ✉, Mark F. Somers[2], Aisling C. Stewart[1], Yosef Alkoby[1], Thomas J. D. Carter[1], Dagmar Butkovicova [1] & Gil Alexandrowicz [1] ✉

Rotational motion lies at the heart of intermolecular, molecule-surface chemistry and cold molecule science, motivating the development of methods to excite and de-excite rotations. Existing schemes involve perturbing the molecules with photons or electrons which supply or remove energy comparable to the rotational level spacing. Here, we study the possibility of de-exciting the molecular rotation of a $D_2$ molecule, from $J = 2$ to the non-rotating $J = 0$ state, without using an energy-matched perturbation. We show that passing the beam through a 1 m long magnetic field, which splits the rotational projection states by only $10^{-12}$ eV, can change the probability that a molecule-surface collision will stop a molecule from rotating and lose rotational energy which is 9 orders larger than that of the magnetic manipulation. Calculations confirm that different rotational orientations have different de-excitation probabilities but underestimate rotational flips ($\Delta m_J \neq 0$), highlighting the importance of the results as a sensitive benchmark for further developing theoretical models of molecule-surface interactions.

[1] Department of Chemistry, Faculty of Science and Engineering, Swansea University, Swansea SA2 8PP, UK. [2] Leiden Institute of Chemistry, Gorlaeus Laboratories, Leiden University, PO Box 9502, 2300 RA Leiden, The Netherlands. ✉email: h.j.chadwick@swansea.ac.uk; g.n.alexandrowicz@swansea.ac.uk

Molecular rotations play a role in a huge range of chemistry related research fields and applications, stimulating the development of experimental techniques to control rotational state populations[1,2]. A common perception, manifested in existing experimental methods, is that to excite, control the directionality and even to de-excite rotational transitions, the external perturbations need to supply or remove energy from the molecule on the same order of magnitude as the spacing between the two rotational energy levels[3–5].

In this work, we present a different approach to controlling rotations, where the probability that a molecule will stop rotating is altered without exposing it to energetic photons or electrons. We show that magnetic manipulations involving pico-eV energy splitting of rotational projection ($m_J$) levels, performed before a collision with a surface, can significantly alter the probability that a deuterium molecule will stop rotating after the collision, converting 22.7 meV rotational energy into translational energy. The experimental results are compared with density functional theory (DFT) based calculations, which relate the observations to the rotational projection states of the impinging $D_2$ but underestimate rotational de-excitation events of helicopter-like molecules.

Our control methodology makes use of the possibility of exciting or de-exciting rotations during a collision with a surface[2,6,7]. Similarly to collisions in the gas phase, which can change the rotational state of the molecule[8], such events can occur when colliding with a surface and can either involve energy exchange with the surface[9], or an internal conversion between the kinetic and rotational energy of the scattered molecule, leading to the phenomena of rotationally inelastic diffraction (RID), see for example ref. [6,10] and [11]. Here, we make use of RID to monitor the probability that a rotating $D_2$ molecule, which approaches a copper surface in the $J = 2$ rotational level, will scatter from the surface in a non-rotating state ($J = 0$). We then alter the probability of this rotational de-excitation in a coherently controlled fashion, using a small magnetic field which splits the nearly degenerate $m_J$ states of the incoming molecules by about a pico-eV, and changes the stereodynamics of the impinging molecules both in terms of rotational projection populations and in terms of the relative phases of the superposition state. Despite what one might expect, the extreme mismatch of energies, the relatively hot temperatures of both the molecular beam and the surface and the large uncertainties in the kinetic energy of the molecules do not prevent us from controlling the rotational de-excitation.

## Results and discussion

A molecular beam of $D_2$, with a mean incident kinetic energy of 38.8 meV and significant population in the $J = 0$, 1 and 2 rotational levels, was scattered from a single crystal Cu(111) sample stabilised at a temperature of 130 K. The details of the molecular interferometry experimental setup we use were published previously[12–14]. Figure 1 shows a schematic of our measurement and the control scheme, briefly described below.

The main elements of the set-up are a hexapole magnet[15] which creates an initial bias in the population of nuclear spin and rotational projection, $m_I$, $m_J$, states. The $m_I$, $m_J$, states which emerge from the hexapole can be considered initially as pure states due to the strong magnetic gradients in this region[16]. A homogenous magnetic field with integral $B1$, produced by an electromagnet, splits the different $m_I$, $m_J$ states, leading to Rabi oscillations of the complex amplitudes of the molecular superposition state. For the case of $J = 2$ molecules, the superposition state can be expressed using a basis set of 30 elements related to the five different nuclear spin projections and five rotational projections of $I = 2$, and an additional five rotational projections

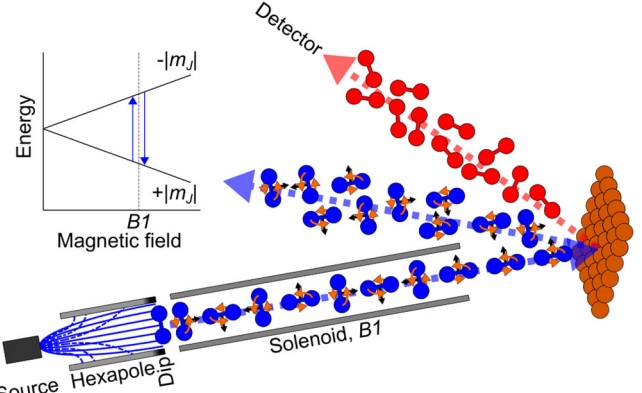

**Fig. 1 Schematic of the coherent magnetic manipulation apparatus.** Overview of the experimental set up showing the positions of the different magnetic field elements used for controlling and manipulating the rotational orientation of the $D_2$ molecules that collide with the Cu(111) surface. The $J = 2$ molecules are shown in blue, and the $J = 0$ in red.

of $I = 0$. If the magnetic Hamiltonian, magnetic field profile and the velocity of the molecules are known, the complex amplitudes of the molecular wave function at the end of the field region and prior to the collision with the surface can be calculated and controlled.

After colliding with the surface, the molecular beam is scattered towards various diffraction channels, each of which reflects the condition for constructive interference of the molecular wave function with itself, for a given incident and scattered molecular velocity and the reciprocal lattice vector of the ordered surface. The phenomena of RID means that one of these diffraction channels is uniquely related to $D_2$ molecules, which approached the surface in a $J = 2$ state and scattered in $J = 0$. If we orient the sample such that this diffraction channel points towards our particle detector[17], the particle count rate becomes a measure of the probability of a molecule undergoing rotational de-excitation. Monitoring that rate as a function of the applied magnetic field integral, $B1$, allows us to measure any change in the de-excitation probability related to our magnetic manipulation scheme, i.e., the molecular quantum state just before it strikes the surface.

Figure 2a presents the scattered signal intensity as a function of incident angle ($\theta_i$) along the [1 0 −1] crystal azimuth. Three peaks are observed in the signal due to different rotational transitions satisfying the condition for diffractive scattering at different incident angles given by

$$n\mathbf{G_x} + m\mathbf{G_y} = \mathbf{k_f}\sin(\theta_t - \theta_i) - \mathbf{k_i}\sin\theta_i \quad (1)$$

where $\mathbf{G_x}$ and $\mathbf{G_y}$ are the reciprocal lattice vectors along the *x* and *y* directions, *n* and *m* the order of diffraction with respect to these two axes (and are both 0 for the three peaks in Fig. 2a), $\theta_t$ the total scattering angle and $\mathbf{k_i}$ and $\mathbf{k_f}$ are the wavevectors before and after scattering, respectively. These are related through conservation of energy via

$$\frac{\hbar^2|\mathbf{k_i}|^2}{2M} - \frac{\hbar^2|\mathbf{k_f}|^2}{2M} = \triangle E_{rot} = BJ'(J' + 1) - BJ(J + 1) \quad (2)$$

where $M$ is the mass and $B$ the rotational constant of $D_2$, $J$ the initial rotational state and $J'$ the final rotational state. For $D_2$, only $\triangle J = 2$ transitions are possible due to nuclear spin statistics associated with the two $I_D = 1$ D atoms (see Supplementary Fig. 1 for a depiction of the possible nuclear spin and low energy rotational states of a $D_2$ molecule). The dominant specular peak (red line) is for rotationally elastic scattering events, whereas the two smaller RID peaks correspond to specific $J$ to $J'$ transitions.

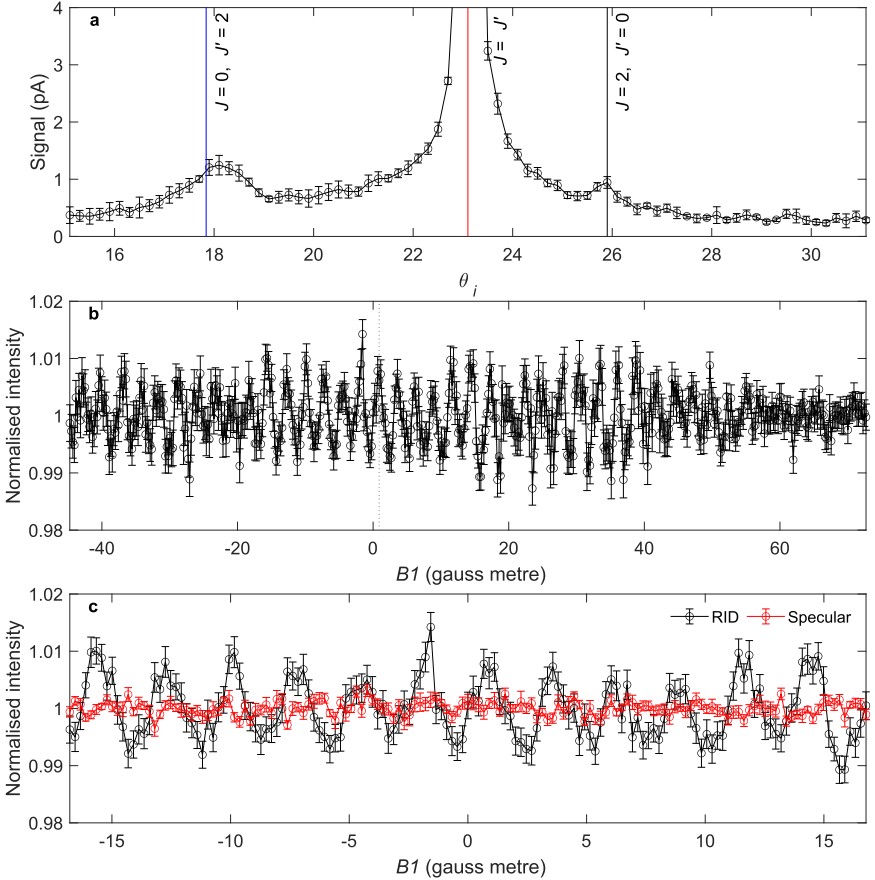

**Fig. 2 Experimental data for D₂ scattering from Cu(111). a** Incident angle scan, showing the positions of the specular scattering peaks for rotationally elastic scattering ($\Delta J = 0$, red) and rotationally inelastic scattering for $J = 0$ to $J' = 2$ (blue) and $J = 2$ to $J' = 0$ (black) transitions. **b** Normalised intensity of D₂ scattering from Cu(111) for the RID peak corresponding to the $J = 2$ to $J' = 0$ transition. The error bars represent standard errors from repeated B1 scans. **c** Comparison of the normalised intensity of D₂ scattering from Cu(111) for the RID peak (black) and the elastic specular peak (red). The error bars represent standard errors from repeated B1 scans. Source data are provided as a Source Data file.

We will focus on the peak seen at an angle of 25.9° degrees, which corresponds to the $J = 2$ to $J' = 0$ transition and $n = m = 0$.

The black circles in Fig. 2b follow the intensity of the $J = 2$ to $J' = 0$ transition while scanning the magnitude of B1. A small but clear oscillation pattern can be seen in the signal, showing that the probability of stopping the D₂ rotation depends on the quantum state of the impinging molecules, which is controlled by our ultra-low-energy magnetic manipulation scheme. The dotted vertical line illustrates that applying a field integral as small as 0.9 gauss metre flips the probability from minimum to maximum. Given the length of the B1 field (~1 m) this corresponds to a field of 0.9 gauss in Fig. 3a, b and a pico-eV splitting of energy (grey dotted line, Fig. 3b).

The $J = 2$ to $J' = 0$ RID peak is ~3° away from the elastic specular peak, where the signal from the base of the elastic specular peak is still significant, as shown in Fig. 2a. To avoid any residual contributions from elastic scattering events, the scattered molecules were passed through a relatively large spoiler magnetic field[12], with a field integral value of 220 gauss metre, where the mismatch in field values was chosen to avoid multiple molecular echoes[13]. To confirm the spoiler field effectively suppresses the elastic contributions and that the oscillations we observe are solely due to rotationally inelastic events, measurements of the signal as a function of B1 was also performed on the specular peak, the result of which (red) is compared with the RID measurement (black) in Fig. 2c. The two measurements were performed using the same magnetic field integral values and

demonstrate that the oscillations we observe are due to the RID scattering and not the elastic specular scattering. Fourier transforms of the two signals are presented in Supplementary Fig. 2, and further demonstrate that the spoiler results in negligible oscillation intensities of the elastically scattered signal, at the relevant frequencies.

To understand our observation we now consider the magnetic manipulation we are performing. Two spin isomers can contribute to the $J = 2$ to $J' = 0$ RID peak, molecules approaching with a total nuclear spin of 0 or 2. The eigen-energies of the 30 states were calculated using the Hamiltonian given in ref. [18] and [19] (see Supplementary Note 2), and are presented in Fig. 3a, b. We have extended our analysis codes, originally developed for $I = 1$, $J = 1$ hydrogen molecules[14], to allow us to propagate each of the initial pure states through the magnetic profile of the beamline. We can then calculate properties such as the scattered signal intensity or the population in a particular $m_I$, $m_J$ state by summing the signals from different propagated wave functions using relative weights that correspond to the transmittance of the initial states through the hexapole and the distribution of velocities within the beam. Figure 3c shows the calculated relative populations for the different $m_J$ states at the position where the beam collides with the surface. The values for each $m_J$ state were obtained by summing over all the $m_I$ states since Cu(111) is non-magnetic, and we expect the nuclear spin to be a spectator to the collision. For an isotropic molecular beam, the fractional population in each $m_J$ state would be 0.2, and the manipulation we perform modulates that value by ~10%.

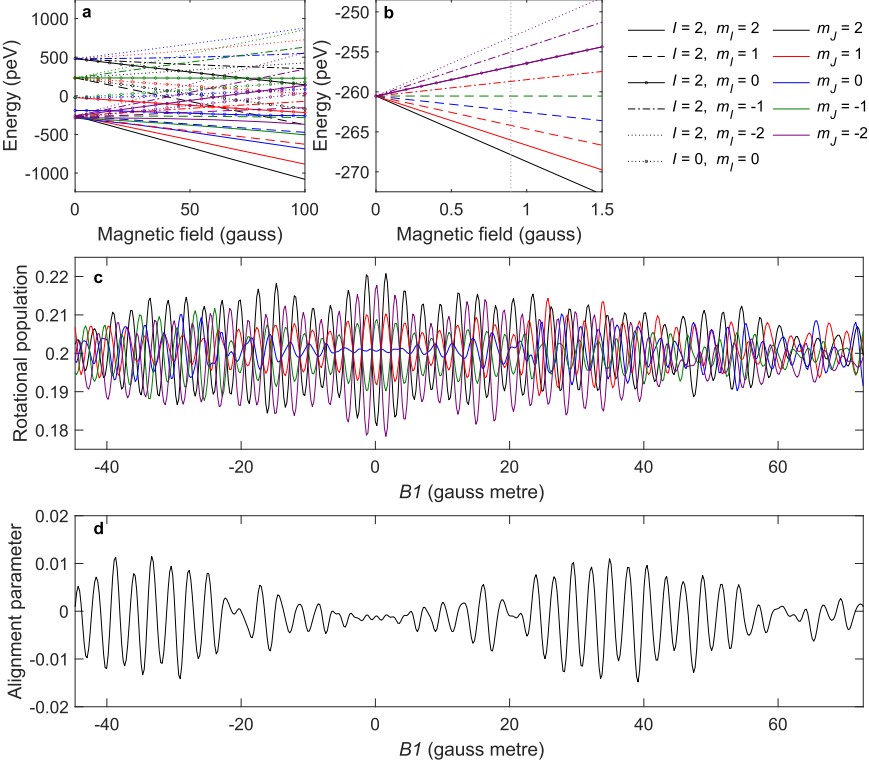

**Fig. 3 Manipulation of the $m_I$, $m_J$ states of the $J = 2$ rotational level of $D_2$ in magnetic fields. a** Magnetic field dependence of the energy of the 30 $m_I$, $m_J$ states for $D_2$ in the $J = 2$ state. The legend specifies the colour scheme used, where different types of lines were used to distinguish different $I$ and $m_I$ combinations, and the colours identify the $m_J$ projection state. **b** Magnification of **a** showing the energy of some of the states in low magnetic field. **c** Calculated populations in the $m_J$ states of the $J = 2$ $D_2$ molecules that collide with the surface. The quantisation axis is the surface normal, and the colour scheme for the $m_J$ populations follows that of **a** and **b**. **d** The alignment parameter of the impinging molecules. Source data are provided as a Source Data file.

Knowing the populations in each $m_J$ state also allows us to calculate the molecular alignment parameter, $a_0^2 = \frac{1}{2} \left\langle \frac{3m_J^2 - J(J+1)}{J(J+1)} \right\rangle$[20], which is shown in Fig. 3d. Similarly to our experimental signal, the alignment oscillates as a function of $B1$, i.e. the relative populations of helicopter and cartwheel like molecules striking the surface are altered. However, there are clear discrepancies between the oscillation of the alignment (Fig. 3d) and the oscillation of the experimentally measured signal related to the rotational de-excitation probability (Fig. 2b), discrepancies which are most noticeable at low values of $|B1|$. This demonstrates that the alignment of the molecules, which is related to the $m_J$ state populations, is not the only factor driving the modulation of the rotational de-excitation probability, and the results cannot be interpreted using a classical picture or simple intuitive principles. This can be expected as the rotational de-excitation event which we are following is a quantum interference phenomena, similar to what was recently demonstrated for gas-phase collision de-excitation[21]. Since our experimental scheme controls the coherent evolution of the states through the apparatus, both the magnitudes and the relative phases of the superposition quantum state of the molecules striking the surface change, and this needs to be taken into account.

To be able to calculate the signal intensity and compare it with the measured signal, we require information about how the wave function of the molecules change when they scatter from the surface. This is characterised by a scattering matrix (S-matrix) relating the complex amplitudes of the incoming rotational wave function to those of the scattered one[22]. The complex elements of the S-matrix were obtained from quantum dynamics calculations using a model[23] in which $D_2$ scatters from an ideal, static Cu(111)

surface within the Born–Oppenheimer approximation. The potential energy surface was computed by DFT employing the optPBE-vdW exchange correlation functional that gives an approximately correct description of the attractive van der Waals interaction[24] and accurately describes reactive scattering of $H_2$ from Cu(111)[25]. Computational details regarding the quantum dynamics simulations can be found in Supplementary Table 1.

The signal calculated using the S-matrix obtained from the DFT calculations is compared with the experimental signal in Fig. 4a. Here, the simulated signal has been scaled so that the maximum amplitude is the same as the experimental data due to an unknown constant background component in the measurement. The calculated signal is characterised by an oscillatory pattern, confirming that the alterations to the spin-rotational parts of the wave functions produced by $B1$ are expected to change the probability of rotational de-excitation of the scattered molecules. A detailed comparison of the two signals (Fig. 4a) shows that while some parts of the signal match quite well, others, especially at low $|B1|$ values, differ by more than the experimental noise.

To understand the origin of the discrepancy between the DFT based calculations and the data, we look at the calculated S-matrix elements shown in Table 1, which predict that only the $m_J = 0$ state undergoes de-excitation with a significant probability, in line with a $\Delta m_J = 0$ propensity rule. This prediction has also been seen in previous calculations of RID transitions on square lattices[26,27]. It is important to note that if there is only one non-zero element in the S-matrix, the shape of the corresponding signal becomes independent of the specific magnitude and phase of the non-zero element. i.e., the red curve in Fig. 4a will be obtained for any S-matrix dominated by the $\Delta m_J = 0$ transition.

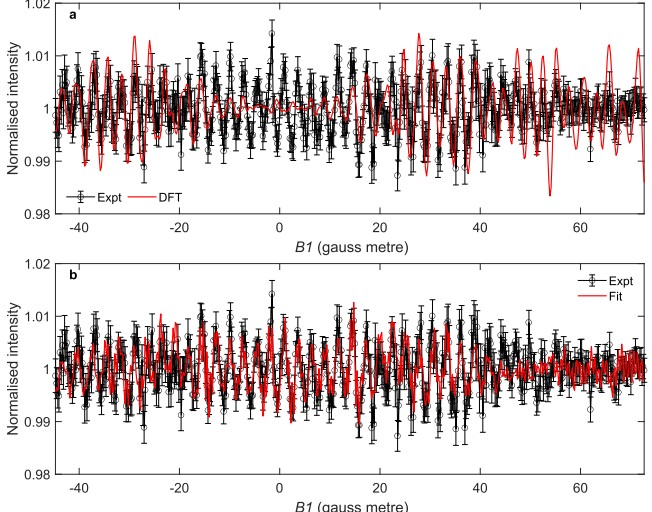

**Fig. 4 Comparison of the experimental data with different models.**
**a** Comparison of the experimentally measured signal for the $J = 2$ to $J' = 0$ RID peak (black) and the signal calculated using the S-matrix obtained from DFT calculations (red). The error bars represent standard errors from repeated $B1$ scans. **b** Comparison of the experimental data (black) with the simulated signal using the best fit S-matrix (red). The error bars represent standard errors from repeated $B1$ scans. Source data are provided as a Source Data file.

**Table 1 S-matrix parameters from the DFT calculations and fit to the experimental data.**

| S-matrix element | DFT | Fit |
|---|---|---|
| $s_{02}/s_{00}$ | 0.02 | 3.42 |
| $s_{01}/s_{00}$ | 0.01 | 1.61 |
| $s_{00}/s_{00}$ | 1.00 | 1.00 |
| $s_{0-1}/s_{00}$ | 0.01 | 1.61 |
| $s_{0-2}/s_{00}$ | 0.03 | 3.42 |
| $k_{02}-k_{00}$ | 2.09 | 5.78 |
| $k_{01}-k_{00}$ | 5.49 | 2.43 |
| $k_{00}-k_{00}$ | 0.00 | 0.00 |
| $k_{0-1}-k_{00}$ | 2.55 | 5.71 |
| $k_{0-2}-k_{00}$ | 2.02 | 5.78 |

Relative changes of the magnitudes ($s_{fn}$) and phases ($k_{fn}$) of the S-matrix elements for scattering from an initial ($n$) $m_J$ state in $J = 2$ to the final ($f$) $m_{J'} = 0$ state in $J' = 0$ from the DFT calculations and from the best fit to the experimental data.

The differences between the pattern of the DFT based signal and measured results illustrate the breakdown of this propensity rule. Previous theoretical work has related a breakdown of this propensity rule to the influence of reactive channels[28]; however, the low energy of the beam and the relatively high dissociation barriers for this surface seem to point to a different mechanism for $\Delta m_J \neq 0$ transitions.

We can also search for S-matrix values that produce a signal which fits the experiment much better, as shown in Fig. 4b and Table 1. It is important to note that, unlike elastic scattering experiments[14], the parameters we obtain from the fitting algorithm are not unique, as we are measuring the flux of the scattered molecules and not performing any further coherent manipulations on the scattered beam. Consequently, we should treat the best fit S-matrix values in Table 1 as an example which could explain the data rather than definitive values. Nevertheless, the improved fit compared to the DFT based signal further

illustrates that to produce the correct oscillation pattern, significant intensity from the $\Delta m_J \neq 0$ elements is needed, meaning the Cu(111) surface must be capable of stopping helicopter-like rotations.

In conclusion, we show in this study that the probability that a $D_2$ molecule will stop rotating after colliding with a surface, can be altered by an extremely subtle magnetic manipulation. The magnetic field we apply creates a pico-eV splitting of the quantum states and coherently controls the spin-rotation wave functions of the molecules which strike the surface. These controlled changes in the molecular wave function alter the probability of rotational de-excitation, a process which involves losing meV of rotational energy, an energy mismatch of nine orders of magnitude with respect to the control energy. The fact that coherent manipulations of the $m_J$ quantum states, alter the de-excitation probability, is in line with various previous results for gas phase, gas-surface and even gas-liquid collisions, which demonstrate the impact of stereodynamics on molecular collisions, see for example ref. [21,29–34].

The experimental result was compared to simulated signals. When using scattering matrix values which were calculated using a DFT based $D_2$-Cu(111) interaction potential, the general phenomena, i.e., a modulation of the de-excitation probability due to the changes in the rotational projection states, is reproduced. However, a detailed comparison reveals discrepancies with the experiment, which are related to an overestimation of the $\Delta m_J \neq 0$ propensity rule, i.e., the theory underestimates the probability that helicopter-like molecules will undergo rotational de-excitation. The discrepancy over $\Delta m_J \neq 0$ transitions could potentially be related to the limitations of the Born–Oppenheimer static surface (BOSS) approximation. The precise role of the lattice dynamics, and how they would affect the discrepancy between the theoretically 0 K calculations presented here and the actual experimentally obtained diffraction data at a surface temperature of 130 K is not fully clear yet. While the effect of lattice dynamics on dissociation and energy exchange has been studied[35–37]. how this would affect the potentially more sensitive rotationally inelastic diffraction is still unknown. Non-adiabatic electronic effects have been shown to impact vibrational excitation[38] but have a negligible effect on dissociation probabilities[39]. An indication that the BOSS approximation might still be valid for the diffraction calculations comes from the case of hydrogen scattering from Pt(111), where BOSS based calculations nicely reproduced experimentally measured trends of diffraction intensities[40].

The quantum state resolved measurements of the type presented here provide a challenging experimental benchmark for further improving molecule-surface interaction models. As the only unknown when simulating the experiments are the S-matrix elements, the accuracy of theoretical models can be tested by their ability to fully reproduce the oscillation patterns of the experimental signal.

A further potential outcome of this work is related to designing scattering based rotational polarising devices for gasphase and gas-surface experiments. The ability to extract and verify S-matrix elements for elastic diffractive scattering channels[14] and now also for inelastic diffraction channels, makes it possible to characterise scattered beams with a pure $J$ state, in terms of their $m_J$ populations. Molecules scattered into these diffraction channels can then be used as a rotationally polarised molecular beam source. Finally, we note that the ultralow energy magnetic manipulations we used here to control and study the de-excitation process rely on the existence of nuclear spin and rotational magnetic moments, making them applicable to various other small molecules in their electronic and vibrational ground states.

## Methods

**Experimental methods**. The experimental apparatus[12–14] used in the present study is described below. A molecular beam of $D_2$ was created using a supersonic expansion through a nozzle held at a temperature of 150 K. The molecules then pass through a skimmer which selects the central part of the beam, producing a translationally cold molecular beam with population in the $J = 0$, 1 and 2 rotational states. From the position of the rotationally inelastic diffraction (RID) peaks in Fig. 2a, it was determined that this gave an incident kinetic energy of 38.8 meV.

The molecules then enter an inhomogeneous magnetic field created by six Halbach-type hexapole magnets[15] where they are either focussed or defocussed depending on whether they are in a low-field seeking or high-field seeking state respectively. After the hexapole, there is a hexapole to dipole transition which adiabatically rotates the projection states to the direction of the dipole, which defines the quantisation axis for the first arm of the machine. This results in the different nuclear spin projection ($m_I$) and rotational projection ($m_J$) states of the $J = 2$ rotational level having different initial populations. Each initial state can be described as a pure state at this position of the apparatus due to the strong magnetic field gradients[16].

After the dipole, the rotational projection states evolve coherently through the beamline before the collision with the surface. This consists of regions of zero magnetic field, regions with small permanent magnetic fields, due to the non-ideality of the apparatus, and a homogeneous tuneable magnetic field with integral, $B1$, created by a solenoid. The solenoid is enclosed by a triple layer of mu-metal shielding, and the currents driving the solenoid are controlled using a Danfysik power supply which can stabilise currents with a ppm accuracy in the range between 0 A and 10 A.

The Cu(111) surface (Surface Preparation Laboratory, The Netherlands) is held on a home built 6-axis manipulator. It was cleaned by heating in an $O_2$ environment (pressure $= 3 \times 10^{-6}$ mbar, temperature $= 300$ K), then $H_2$ (pressure $= 5 \times 10^{-7}$ mbar, temperature $= 600$ K) followed by flashing to a temperature of 700 K. The quality and structure of the sample was verified using LEED and helium scattering. During the measurements presented here, it was held at a temperature of 130 K.

After the collision with the surface, the molecules pass down the second arm of the apparatus, which is at an angle of 46.2° to the first. The beamline in this second arm consists of a second solenoid and a second hexapole magnet. Due to the velocity of the molecules in the second arm ($>1700$ ms$^{-1}$), the state selection in the second hexapole will not be significant, meaning that the second arm can simply be considered to act as a high angular resolution filter. To ensure this is the case, we also passed a relatively large current through the second solenoid which acts as a spoiler field, removing any residual coherencies which might exist in the scattered beam from elastic scattering events. At the end of this arm is a high-efficiency particle detector[17] which measures the flux of the scattered molecules.

**Data analysis**. The extraction of the S-matrix elements from the experimental signal is performed by minimising the difference between the measured data and calculated signal. This is achieved using the downhill simplex method of Nelder and Mead[41] in combination with a simulated annealing algorithm. The minimisation is repeated 100 times with random initial guesses for the amplitudes and phases of the S-matrix elements to try and obtain results that correspond to the global minimum as opposed to a local minimum. The parameters that are presented in Table 1 were obtained from the six best results out of 100 fits.

**Signal calculation**. The method for calculating the signal for a specific scattering matrix follows a similar procedure to that used to interpret the measurements of $H_2$ elastically scattering from a LiF surface[14], but extended to a 30 level case rather than the nine level case of $H_2$ in $I = 1$, $J = 1$. As only the $I = 0$, $J = 2$ and $I = 2$, $J = 2$ initial states can contribute to the RID signals that we are calculating, we simulate the propagation of these initial states through the first arm of the machine before the scattering takes place. The initial states are weighted according to their populations in an isotropic molecular beam expansion, i.e., the ratio $I = 2$: $I = 0$ is 5:1.

The probability that each $m_I$, $m_J$ projection state is transmitted through the first hexapole is determined using semi-classical trajectory calculations[42]. As the magnetic field gradients are large, the superposition states decohere[16] meaning the initial $m_I$, $m_J$ states ($n$) can be described as pure states, just with different populations (related to the transmission probabilities down the first hexapole, $P_{hex1}(n)$). The quantisation axis is chosen to coincide with the direction of the dipole field at the end of the hexapole lens, which we define as the $Z$-axis.

The propagation of the initial states through the rest of the machine is done using the Hamiltonian

$$\mathcal{H}(\mathbf{B}) = \frac{\hbar^2 \mathbf{k}^2}{2M} + \mathcal{H}_R(\mathbf{B}) \tag{3}$$

The first term corresponds to the motion of the centre of mass which is treated classically in the propagation, and the second term accounts for the quantum mechanical evolution of the rotational and nuclear spin projection states (see Supplementary Note 2). The classical treatment of the first term was shown to be sufficiently accurate as long as the magnetic field is not very large[43].

In the calculation, each initial $m_I$, $m_J$ state is propagated coherently through the measured magnetic field profile of the first arm, producing a propagation matrix $U(B1)$. In an ideal apparatus this profile would contain only the magnetic fields of the dipole which defines the quantisation axis, and the first solenoid, $B1$. However, there are small residual fields in the first arm of the machine, which cause some additional mixing of the states, and are also included in the calculation.

The quantisation axis of the scattering matrix ($S$) is taken to be along the surface normal ($Z_N$), whereas the propagation down the first arm is taken to be with respect to the $Z$-axis. It is therefore necessary to rotate the reference frame from $Z$ to $Z_N$ using a rotation matrix $R(\theta_1)$ where $\theta_1$ depends on the incident angle, $\theta_i$. $S$ then acts on this rotated wave function, where it is assumed that the five initial $m_I$ states of each $m_J$ state of $I = 2$, $J = 2$, and the $m_I = 0$ state of each $m_J$ state for $I = 0$, $J = 2$, all scatter the same, i.e., the nuclear spin is a spectator to the collision. It follows that $S$ can be written as

$$S = \begin{pmatrix} s_{02}e^{ik_{02}} & s_{01}e^{ik_{01}} & s_{00}e^{ik_{00}} & s_{0-1}e^{ik_{0-1}} & s_{0-2}e^{ik_{0-2}} \end{pmatrix} \tag{4}$$

where $s_{fn}$ is the magnitude associated with scattering from $n$ to $f$ and $k_{fn}$ the corresponding phase.

Due to the velocity of the molecules through the second arm of the machine and the molecules being in the $J = 0$ state, there is no need to calculate the evolution of the projection states through the second arm, and the signal is taken simply as proportional to the scattered flux into the second arm, i.e., the square of the outgoing wave function. The wave function for a molecule initially in state $n$ which reaches the detector can therefore be calculated as

$$|\psi_{fn}^{Z_N}\rangle = SR(\theta_1)U(B1)\sqrt{P_{hex1}(n)}|n\rangle \tag{5}$$

from which the signal intensity (sig) can be calculated as

$$\text{sig} = \sum_v P_v \sum_f \sum_n \left\langle \psi_{fn}^{Z_N} | \psi_{fn}^{Z_N} \right\rangle \tag{6}$$

where the sums run over the initial states, final states and the velocity distribution of the molecular beam expansion which is modelled as a gaussian to give the velocity weights $P_v$.

## Data availability

Source data for the figures are provided with this paper. The propagation matrices which give the wave function elements for each experimental condition are available from the corresponding authors on request. Due to the unique nature of the outputs, guidance would be needed from the corresponding authors who would be happy to provide such guidance and full access to the data. Source data are provided with this paper.

## Code availability

The codes used to calculate the wave function matrix elements and the simulated signal can be obtained from the corresponding authors, which will also provide guidance on how to use them.

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

## Acknowledgements
The authors would like to thank Prof. Geert-Jan Kroes for stimulating scientific discussions and Prof. Mark Brouard for stimulating scientific discussions and critical reading of the manuscript. This work was funded by an ERC consolidator grant (Horizon 2020 Research and Innovation Programme grant 772228) (G. A.) and an EPSRC New Horizons grant (EP/V048589/1) (G. A.).

## Author contributions
G. A. and H. C. conceived and supervised the project. H. C, A. C. S., T. J. D. C. and D. B. performed the experiments. Y. A. installed and characterised the experimental apparatus. H. C. developed and performed the data analysis. M. F. S. performed the DFT calculations. G. A. and H. C. wrote the paper.

## Competing interests
The authors declare no competing interests.
