## [Peer Review File · Nature Communications]

REVIEWER COMMENTS

Reviewer #1 (Remarks to the Author):

This is a nicely designed experiment from one of the leading experimental groups in magnetic manipulation of molecules scattering from solid surfaces. The authors show in this work how to realize rotational deexcitation of a D₂ molecule upon collision with a metal surface, without resorting to photonic or electronic excitation. This is made possible by counting the molecules scatter into the rotationally inelastic diffraction ($J=2 \rightarrow 0$) channel. The probability of this rotational de-excitation is coherently controlled by using a small magnetic field which split the m_j states and changes m_j state populations of the D₂ molecules and thus the overall stereodynamics of the scattering process. To explain the mechanism, they also perform quantum scattering calculations based on a six-dimensional DFT-interpolated PES, although the agreement with experiment is less satisfactory. This indicates the necessity of further development of theoretical models of molecule-surface interactions. While I am impressed by the novel technique and do like the present story, I think that the current manuscript can be improved for provide more valuable insights and clear evidence on the significance of this work. I have the following concerns for the authors to improve the manuscript.

1. Regarding the significance of this work, the authors choose to emphasize that they propose an alternative way of controlling molecular rotation by the possibility of exciting or de-exciting rotations of the molecule possible during a collision with a surface. But isn't possible in every molecule-surface scattering system as long as the rotationally inelastic channel exists? Such scattering or diffraction experiments have been conducted for years. It is true that the present experiment is unique in manipulating the initial magnetic states of the molecule and detecting the particular diffraction channel of the $J=0$ state at a specific angle. How could one make use of the scattered molecules in a specific rotational state? In addition, the experiment shows the possibility of the diffraction signal of $J=0$ which has only one m_j state. Is it possible to use this technique to select one of the final m_j states for $J>0$? These questions are important to give an id general picture for the broad readership in Nat. Commun.

2. From Table 1, it is clear that calculated results are qualitatively inconsistent with experimental ones. The former contain almost exclusively the $m_j=0 \rightarrow 0$ transition, which is however the smallest component in the latter. The authors quote the previous theoretical work for H₂+Co(0001) for a possible interpretation. However, in my opinion, the influence of reactive channels is unlikely in effect in D₂+Cu(111), given the very low translational incidence energy here (~38 meV). Note that the late dissociation barrier on Cu(111) is ~0.6 eV, sufficiently high that the molecule can hardly access the barrier region at all during inelastic scattering. In comparison, the barrier is much lower and earlier on Co(0001) (~0.05 eV), where the reactive channel could play a role in transitions between different m_j states. These results need a more reasonable explanation.

3. Preferably, the authors may suggest which type of information of the molecule-surface interaction needs to be improved so as to better reproduce the oscillation patterns of the experimental signal.

In my personal point of view, the $\text{detal}(m_j)=0$ propensity rule seems to be intrinsic and largely insensitive to the density functional used (unless the authors can provide some evidence), in the BOSS approximation. Could the lattice motion or anisotropic/ mode specific electronic friction induced nonadiabatic effect play a role? These insights (even guess) will help guide theorists to develop more advanced models.

Reviewer #2 (Remarks to the Author):

Review of "Stopping molecular rotation using coherent ultra-low-energy magnetic manipulations"

The authors have demonstrated that magnetic manipulations involving the pico-eV energy splitting of magnetic sublevels of a rovibrational level, performed before a collision with a surface, can significantly alter the probability that a deuterium molecule will stop rotating after the collision, converting 22.7 meV rotational energy into translational energy. I regard this finding as a most excellent result, but I also must confess that I am not surprised by this fact. To me, this work is an example of stereodynamics in molecular collisions. Please take a look at

"Quantum Stereodynamics of H₂ Scattering from Co(0001): Influence of Reaction Channels" by Marcos del Cueto, Xueyao Zhou, Alberto S. Muzas, Cristina Díaz, Fernando Martín, Bin Jiang, and Hua Guo, *J. Phys. Chem. C* 2019, 123, 16223–16231.

and

"Stereodynamics in state-resolved scattering at the gas–liquid interface" by Bradford G. Perkins, Jr. and David J. Nesbitt, *Proc. Natl. Acad. Sci. (USA)* 2008 105, 12684–12689.

There are many more references (and this reviewer is not one of the authors cited above). I cannot recommend publication until this work is put into the context of what has been done before in this field. This should be easy for the authors to accomplish, so I have suggested minor revision prior to publication.

Reviewer #3 (Remarks to the Author):

Referee report on the article by Chadwick et al (NCOMMS-21-49982-T)

The article describes a very sophisticated relatively complicated experiment in which it is demonstrated that it is possible to influence the rotationally inelastic collision of a D₂ molecule from a Cu(111) surface by a 9 orders of magnitude smaller magnetic interaction than the rotational energy transfer in surface collision.

The referee had significant problems in fully understanding the manuscript in a reasonable amount of time. Two reasons are that the theory behind the experiment is extremely complicated and not fully understood. Probably the results are not possible to be fully explained in a letter type communication. The discussion in the conclusions section reflects this where it is written “ The quantum state resolved measurements of the type presented here provide a challenging experimental benchmark for further improving molecule-surface interaction models to the point where the S-matrix elements calculated from these models will be capable of fully reproducing the oscillation patterns of the experimental signal”

The second reason is that in the opinion of the referee the article is not well written. In the following the referee gives several examples where he was led astray and confused.

The claim in the abstract that a magnetic interaction of 9 orders smaller than the energy transfer is sufficient to affect an inelastic collision is an incomplete description of the experiment. In actual fact the paper describes an experiment in which the phase of the wave packet passing through a long solenoid is affected by a field of 0.9 gauss meter, where the effect also depends on the length of the solenoid. This is finally said on page 4 at the end of the first paragraph in the sentence beginning with “Monitoring that rate.....” If expressed in such a way in the abstract it would not be misleading completely understandable to most readers of Nature Comm.

Furthermore the figures are not well presented and the captions are inadequate. The referee missed in the introduction an energy diagram with the rotational and the nuclear spin states of the D₂ molecule. Instead Fig 2a is shown without any explanation of the 4 levels at zero magnetic field neither in the text nor in the caption. Nor was it possible to identify the 9 lines in Fig 3b despite the apparent assignments on the right side.

Fig 2b would have been much clearer if the authors had enlarged only the part at small field strengths which is the important fundamental observation of the paper. An enlargement is then

shown in Fig 2c but there the oscillations seem to be different or possibly this may be apparent because of the different scale. Furthermore the remaining red oscillations are not explained. Is it because the spoiler field was not sufficiently large?

Fig 3d shows the alignment. Why the alignment is zero at the critical field of 0.9 gauss meter is not addressed? In the bottom paragraph of page 6 reference is made to two oscillation patterns. Are they on both sides of $B_1=0$? In that case the difference is hardly apparent and should be shown superimposed since it appears to be an important observation.

The referee was also not able to understand the meaning of Table 1 and why there are large differences in the columns labeled DFT values and Fit values. Is it because as they report on the middle of page 8 “ Consequently, we should treat the best fit S-matrix values in Table 1 as an example which could explain the data.....”?

In trying to better understand the article the referee looked up another Nature Comm, Ref 14 by the same group which he found well written and fully understandable and helped the referee to better understand the article under review.

In summary, the results are interesting but the present manuscript is not suited for a letters journal article. With more detail the results surely would be publishable in a journal devoted to longer articles.

The comments from the Reviewers are reproduced in black below, with our replies written in red. Any changes that have been made to the manuscript are given both here and in the revised manuscript in blue. We thank the Reviewers for their helpful comments, and address each of them in turn below.

Reviewer #1 (Remarks to the Author):

This is a nicely designed experiment from one of the leading experimental groups in magnetic manipulation of molecules scattering from solid surfaces. The authors show in this work how to realize rotational de-excitation of a D2 molecule upon collision with a metal surface, without resorting to photonic or electronic excitation. This is made possible by counting the molecules scatter into the rotationally inelastic diffraction ($J=2 \rightarrow 0$) channel. The probability of this rotational de-excitation is coherently controlled by using a small magnetic field which split the m_j states and changes m_j state populations of the D2 molecules and thus the overall stereodynamics of the scattering process. To explain the mechanism, they also perform quantum scattering calculations based on a six-dimensional DFT-interpolated PES, although the agreement with experiment is less satisfactory. This indicates the necessity of further development of theoretical models of molecule-surface interactions.

While I am impressed by the novel technique and do like the present story, I think that the current manuscript can be improved for provide more valuable insights and clear evidence on the significance of this work. I have the following concerns for the authors to improve the manuscript.

We thank the Reviewer for their supportive comments and address their concerns below.

1. Regarding the significance of this work, the authors choose to emphasize that they propose an alternative way of controlling molecular rotation by the possibility of exciting or de-exciting rotations of the molecule possible during a collision with a surface. But isn't possible in every molecule-surface scattering system as long as the rotationally inelastic channel exists? Such scattering or diffraction experiments have been conducted for years. It is true that the present experiment is unique in manipulating the initial magnetic states of the molecule and detecting the particular diffraction channel of the $J=0$ state at a specific angle. How could one make use of the scattered molecules in a specific rotational state? In addition, the experiment shows the possibility of the diffraction signal of $J=0$ which has only one m_j state. Is it possible to use this technique to select one of the final m_j states for $J>0$? These questions are important to give an id general picture for the broad readership in Nat. Commun.

While more than likely, the most immediate use of our results is to test and improve theoretical modelling of molecule-surface interaction (see the response to further comments below), we agree with the reviewer that further discussion of the potential applications of the control scheme would be of interest to the broad readership of Nature Communications.

In a recent paper [*Nat. Commun.* **11**, 3110 (2020)], we have shown that the magnetic manipulation technique can be used to extract the scattering (S) matrix of a molecule-surface scattering experiment and in particular determine the m_j populations of $J > 0$ states within different **elastic diffraction scattering channels**. This characterisation, i.e., knowing the S-matrix elements, should make it possible to use a simple surface (for example a LiF crystal) as a rotational polariser within gas-phase or gas-surface experimental setups which do not have magnetic manipulation control. Scattering an initially unpolarised beam from this surface and selecting particular diffraction channels as a secondary beam source, would provide rotationally polarised capabilities which were previously out of reach. A limitation of using elastic diffraction to rotationally polarise beams is that while the m_j populations of a specific J state can be altered, there is no J state selection, and other J states which always exist in the beam to a certain amount will also scatter into the elastic diffraction peak, complicating the analysis and degrading the state selectivity. The ability to characterise the rotational projection states in rotationally inelastic diffraction channels with $J_{\text{final}}>0$, adds the missing selectivity, making the scattering

surface a rotational polariser, with control over both J and m_J degrees of freedom. A discussion of this potential application has been added to page 10 of the manuscript:

A further potential outcome of this work is related to designing scattering based rotational polarising devices for gas-phase and gas-surface experiments. The ability to extract and verify S -matrix elements for elastic diffractive scattering channels¹⁴ and now also for inelastic diffraction channels, makes it possible to characterise scattered beams with a pure J state, in terms of their m_J populations. Molecules scattered into these diffraction channels can then be used as a rotationally polarised molecular beam source.

Looking more broadly at quantum control schemes of atoms / molecules, the results demonstrate a rather extreme case of a low energy control scheme which is many orders of magnitude lower than both the energy exchange process it controls, and perhaps even more importantly, the energy exchanged by random processes related to the elevated beam and surface temperatures. There is an interesting analogy with the principle of operation of a transistor, in the sense that the large changes in the rotational energies effectively amplify the subtle magnetic perturbation energy used to control the process. However, as we do not have a specific application in mind which exploits the huge energy mismatch, we decided it is far too speculative at this stage to include these thoughts in the manuscript.

2. From Table 1, it is clear that calculated results are qualitatively inconsistent with experimental ones. The former contain almost exclusively the $m_J=0 \rightarrow 0$ transition, which is however the smallest component in the latter. The authors quote the previous theoretical work for $H_2+Co(0001)$ for a possible interpretation. However, in my opinion, the influence of reactive channels is unlikely in effect in $D_2+Cu(111)$, given the very low translational incidence energy here (~ 38 meV). Note that the late dissociation barrier on $Cu(111)$ is ~ 0.6 eV, sufficiently high that the molecule can hardly access the barrier region at all during inelastic scattering. In comparison, the barrier is much lower and earlier on $Co(0001)$ (~ 0.05 eV), where the reactive channel could play a role in transitions between different m_J states. These results need a more reasonable explanation.

We agree with the reviewer on this point, we did not intend to say that reactive channels are necessarily the relevant reason for $\Delta m_J \neq 0$ transitions in this system, but our text was definitely not clear enough. The text has been changed to emphasize the point stressed by the referee by adding the following sentence on page 8:

however, the low energy of the beam and the relatively high dissociation barriers for this surface, seem to point to a different mechanism for $\Delta m_J \neq 0$ transitions.

3. Preferably, the authors may suggest which type of information of the molecule-surface interaction needs to be improved so as to better reproduce the oscillation patterns of the experimental signal. In my personal point of view, the $d_{\text{et}}(m_J)=0$ propensity rule seems to be intrinsic and largely insensitive to the density functional used (unless the authors can provide some evidence), in the BOSS approximation. Could the lattice motion or anisotropic/ mode specific electronic friction induced nonadiabatic effect play a role? These insights (even guess) will help guide theorists to develop more advanced models.

We agree with the referee that lattice motion or mode-specific electronic friction could potentially contribute to the breakdown of the propensity rule, and while this effect might be small, it is appropriate to mention this in the paper. The role of electronic friction and lattice dynamics have been studied for hydrogen colliding with $Cu(111)$ [J. Phys. Chem. Lett. 9, 1803–1808 (2018), J. Chem. Phys. 137, 054703 (2012), J. Chem. Phys. 149, 234702 (2018) and J. Chem. Phys. 154, 74710 (2021)], these studies followed properties such as dissociation probability and energy exchange rather than low energy diffractive scattering.

The mentioned propensity rule is understood for perfect crystal structures, but for thermally distorted surfaces this could be different. Earlier work on using the Static Corrugation Method

(SCM) has shown that the rotational quadrupole alignment for reaction is lowered a little by the increased surface temperature (J. Chem. Phys. **137**, 054703 (2012)), however, for this experiment the surface temperature is only 130 K and the effects could be quite small. Another study using an improved SCM showed that rovibrational inelastic scattering can also be affected by the introduced (static) corrugation of an elevated surface temperature, but again for higher surface temperatures than considered here (J. Chem. Phys. **149**, 234702 (2018)). The more sensitive diffraction probabilities, under investigation in this work, could still be somewhat affected for 130 K but currently we are not able to give a precise estimate of how much. An indication that the BOSS approximation might still be valid for calculating diffraction comes from the case of hydrogen scattering from Pt(111), where BOSS based calculations nicely reproduced experimentally measured trends of diffraction intensities (Science, **312**, 86 (2006)). We have added the following text and references to the paper on page 9:

The discrepancy over $\Delta m \neq 0$ transitions could potentially be related to the limitations of the Born Oppenheimer static surface (BOSS) approximation. The precise role of the lattice dynamics, and how they would affect the discrepancy between the theoretically 0K calculations presented here and the actual experimentally obtained diffraction data at a surface temperature of 130K is not fully clear yet. While the effect of lattice dynamics on dissociation and energy exchange has been studied³⁵⁻³⁷, how this would affect the potentially more sensitive rotationally inelastic diffraction is still unknown. Non-adiabatic electronic effects have been shown to impact vibrational excitation but have a negligible effect on dissociation probabilities³⁸. An indication that the BOSS approximation might still be valid for the diffraction calculations comes from the case of hydrogen scattering from Pt(111), where BOSS based calculations nicely reproduced experimentally measured trends of diffraction intensities³⁹.

35. Wijzenbroek, M. & Somers, M. F. Static surface temperature effects on the dissociation of H₂ and D₂ on Cu(111). *J. Chem. Phys.* **137**, 54703 (2012).
36. Spiering, P., Wijzenbroek, M. & Somers, M. F. An improved static corrugation model. *J. Chem. Phys.* **149**, 234702 (2018).
37. Smits, B. & Somers, M. F. Beyond the static corrugation model: Dynamic surfaces with the embedded atom method. *J. Chem. Phys.* **154**, 74710 (2021).
38. Spiering, P. & Meyer, J. Testing Electronic Friction Models: Vibrational De-excitation in Scattering of H₂ and D₂ from Cu(111). *J. Phys. Chem. Lett.* **9**, 1803–1808 (2018).
39. Nieto, P. *et al.* Reactive and nonreactive scattering of H₂ from a metal surface is electronically adiabatic. *Science* **312**, 86–89 (2006).

Reviewer #2 (Remarks to the Author):

Review of "Stopping molecular rotation using coherent ultra-low-energy magnetic manipulations"

The authors have demonstrated that magnetic manipulations involving the pico-eV energy splitting of magnetic sublevels of a rovibrational level, performed before a collision with a surface, can significantly alter the probability that a deuterium molecule will stop rotating after the collision, converting 22.7 meV rotational energy into translational energy. I regard this finding as a most excellent result, but I also must confess that I am not surprised by this fact. To me, this work is an example of stereodynamics in molecular collisions. Please take a look at "Quantum Stereodynamics of H₂ Scattering from Co(0001): Influence of Reaction Channels" by Marcos del Cueto, Xueyao Zhou, Alberto S. Muzas, Cristina Díaz, Fernando Martín, Bin Jiang, and Hua Guo, *J. Phys. Chem. C* 2019, 123, 16223–16231. and "Stereodynamics in state-resolved scattering at the gas–liquid interface" by Bradford G. Perkins, Jr. and David J. Nesbitt, *Proc. Natl. Acad. Sci. (USA)* 2008 105, 12684–12689. There are many more references (and this reviewer is not one of the authors cited above). I cannot recommend publication until this work is put into the context of what has been done before in this field. This should be easy for the authors to accomplish, so I have suggested minor revision prior to publication.

We accept the reviewer's comment that it is important to relate this work to stereodynamics and previous studies which demonstrated the effect of stereodynamics on molecular collisions. We have added / modified the text to address this as follows.

In the abstract we have emphasised that different rotational orientations have different de-excitation probabilities by adding the following text in the abstract:

Calculations confirm that different rotational orientations have different de-excitation probabilities,

We have also added some text and references to examples which demonstrate the role of stereodynamics in molecular collisions on page 9:

The fact that coherent manipulations of the m_J quantum states, alter the de-excitation probability, is in line with various previous results for gas phase, gas-surface and even gas-liquid collisions which demonstrate the impact of stereodynamics on molecular collisions, see for example references ^{21,29–34}.

21. Haowen, Z., E., P. W., Nandini, M. & N., Z. R. Quantum mechanical double slit for molecular scattering. *Science* **374**, 960–964 (2021).
29. Perkins, B. G. & Nesbitt, D. J. Stereodynamics in state-resolved scattering at the gas–liquid interface. *Proc. Natl. Acad. Sci.* **105**, 12684–12689 (2008).
30. Yoder, B. L., Bisson, R. & Beck, R. D. Steric effects in the chemisorption of vibrationally excited methane on Ni(100). *Science* **329**, 553–556 (2010).
31. Hou, H., Gulding, S. J., Rettner, C. T., Wodtke, A. M. & Auerbach, D. J. The stereodynamics of a gas-surface reaction. *Science* **277**, 80–82 (1997).
32. Heid, C. G. *et al.* Controlling the Spin–Orbit Branching Fraction in Molecular Collisions. *J. Phys. Chem. Lett.* **12**, 310–316 (2021).
33. Sharples, T. R. *et al.* Non-intuitive rotational reorientation in collisions of NO(A $^2\Sigma^+$) with Ne from direct measurement of a four-vector correlation. *Nat. Chem.* **10**, 1148–1153 (2018).
34. Onvlee, J. *et al.* Imaging quantum stereodynamics through Fraunhofer scattering of NO radicals with rare-gas atoms. *Nat. Chem.* **9**, 226–233 (2017).

Reviewer #3 (Remarks to the Author):

Referee report on the article by Chadwick et al (NCOMMS-21-49982-T)

The article describes a very sophisticated relatively complicated experiment in which it is demonstrated that it is possible to influence the rotationally inelastic collision of a D2 molecule from a Cu(111) surface by a 9 orders of magnitude smaller magnetic interaction than the rotational energy transfer in surface collision.

The referee had significant problems in fully understanding the manuscript in a reasonable amount of time.

We thank the referee for their sincere remark, and have tried to make the paper more accessible, as described below.

Two reasons are that the theory behind the experiment is extremely complicated and not fully understood.

We strongly disagree with the referee on this point, the theory behind the experiment is completely understood, it involves solving the well-established Ramsey Hamiltonian for a molecule propagating through known and controlled magnetic fields. The theory which isn't fully understood is that of molecule-surface scattering, or more precisely, while the reliability and accuracy of approximations which are needed to solve molecule-surface interactions are

constantly improving, this is still an active research field with considerable challenges ahead, making this work particularly useful as an experimental benchmark.

Probably the results are not possible to be fully explained in a letter type communication. The discussion in the conclusions section reflects this where it is written “ The quantum state resolved measurements of the type presented here provide a challenging experimental benchmark for further improving molecule-surface interaction models to the point where the S-matrix elements calculated from these models will be capable of fully reproducing the oscillation patterns of the experimental signal”

Again the referee is confusing the complexity of the technique which can reliably test any S-matrix prediction, with the fact that that even the best theoretical models are still not capable of accurately calculating these elements. To try and avoid misunderstanding of this point we rephrased the text in page 10 to read:

The quantum state resolved measurements of the type presented here provide a challenging experimental benchmark for further improving molecule-surface interaction models. As the only unknown when simulating the experiments are the S-matrix elements, the accuracy of theoretical models can be tested by their ability to fully reproduce the oscillation patterns of the experimental signal.

The second reason is that in the opinion of the referee the article is not well written. In the following the referee gives several examples where he was led astray and confused.

The claim in the abstract that a magnetic interaction of 9 orders smaller than the energy transfer is sufficient to affect an inelastic collision is an incomplete description of the experiment. In actual fact the paper describes an experiment in which the phase of the wave packet passing through a long solenoid is affected by a field of 0.9 gauss meter, where the effect also depends on the length of the solenoid. This is finally said on page 4 at the end of the first paragraph in the sentence beginning with “Monitoring that rate.....” If expressed in such a way in the abstract it would not be misleading completely understandable to most readers of Nature Comm.

The reviewer is correct that the length of the magnetic field will affect the energy splitting needed, although any physically practical length would still result in a huge energy mismatch. We had already stated this in the original manuscript, but not in the abstract. Following the remark of the reviewer we have altered the text in the abstract to make this point clearer from the start by adding the text highlighted below.

We show that passing the beam through a 1m long magnetic field, which splits the rotational projection states by only 10^{-12} eV, can change the probability that a molecule-surface collision will stop a molecule from rotating and lose rotational energy which is 9 orders larger than that of the magnetic manipulation.

Furthermore the figures are not well presented and the captions are inadequate. The referee missed in the introduction an energy diagram with the rotational and the nuclear spin states of the D₂ molecule. Instead Fig 2a is shown without any explanation of the 4 levels at zero magnetic field neither in the text nor in the caption.

We have added an energy diagram in the Supplementary Information section with a statement added on page 4 in the main manuscript to direct readers to it:

(see Fig. 1 of the Supplementary Information for a depiction of the possible nuclear spin and low energy rotational states of a D₂ molecule)

We have also added a paragraph and a caption to that figure in the Supplementary Information which explains the number of states and their origin.

1. Nuclear spin and rotational states of D₂ molecules

Individual D atoms have a nuclear spin, I_D , of 1, which means that the total nuclear spin of a D₂ molecule, I , can have values of 2, 1 or 0 depending on how the spins of the individual atoms couple. Molecules with even values of I correspond to ortho-D₂, and with odd values of I , to para-D₂. As the D atoms are bosons, the molecular wave-function has to be symmetric with respect to interchange of the two nuclei, which restricts ortho-D₂ to having even rotational states, and para-D₂ to odd rotational states. Each I state has $2I + 1$ nuclear spin projection states (m_I), and each J state has $2J + 1$ rotational projection states (m_J). Fig. 1 illustrates the relation between the total nuclear spin states and the allowed rotational states.

Fig. 1. Schematic representation of the nuclear spin states of D₂ and the lowest rotational states that they can have. The $I = 0, J = 0$ is a singlet, whereas $I = 0, J = 2$ splits into 5 non-degenerate states, $I = 1, J = 1$, into 9 non-degenerate states, $I = 2, J = 0$ into 5 non-degenerate states and $I = 2, J = 2$ into 25 non-degenerate states. The experiments we perform isolate the $J = 2 \rightarrow J = 0$ transition, and therefore involves 30 initial states, the field dependence of the eigen-energies is plotted in Fig. 3a of the main manuscript. Note that for a field of zero there are only 6 different eigen-energies, associated with 6 different hyperfine states¹.

1. Code, R. F. & Ramsey, N. F. Molecular-beam magnetic resonance studies of HD and D₂. *Phys. Rev. A* **4**, 1945–1959 (1971).

Regarding the lack of explanation of the “4 levels”, we assume the referee means the 6 energy level splitting at zero field. This is now noted in the caption of figure 1 (given above) in the Supplementary Information section together with a reference to a paper which explains this in more detail.

Nor was it possible to identify the 9 lines in Fig 3b despite the apparent assignments on the right side.

Indeed it is difficult to follow the many lines in figures 3a and 3b, however, we think this is an inherent problem related to the complexity of the system rather than a particular presentation method. After trying a few alternatives we decided that the existing colour coding / assignments is probably the best way to present the curves, but we have changed the caption to assist the reader in understanding the colour scheme.

The new caption to figure 3 now reads:

Manipulation of the m_I, m_J states of the $J=2$ rotational level of D₂ in magnetic fields. (A) Magnetic field dependence of the energy of the 30 m_I, m_J states for D₂ in the $J=2$ state. The legend specifies the colour scheme used, where different types of lines were used to distinguish different I and m_I combinations and the colours identify the m_J projection state. **(B)** Magnification of Panel A showing the energy of some of the states in low magnetic field. **(C)** Calculated populations in the m_J states of the $J=2$ D₂ molecules that collide with the surface. The quantisation axis is the surface normal and the colour scheme for the m_J populations follows that of panels A and B. **(D)** The alignment parameter of the impinging molecules.

Fig 2b would have been much clearer if the authors had enlarged only the part at small field strengths which is the important fundamental observation of the paper.

We actually disagree with the referee on this point. To test a theoretical prediction of the S matrix, and even more so to try and determine it directly from the data, it is important to measure an oscillation pattern within a magnetic field range which is as wide as possible. This makes the fit (or misfit) when testing S-matrix values more significant. As we scan the field both the amplitude and the phase of the wavefunction changes in a complex way, the probability of a D₂ molecule undergoing the $J = 2 \rightarrow J = 0$ transition changes accordingly, producing the complex oscillatory pattern.

An enlargement is then shown in Fig 2c but there the oscillations seem to be different or possibly this may be apparent because of the different scale. Furthermore the remaining red oscillations are not explained. Is it because the spoiler field was not sufficiently large?

The enlargement in figure 2c plots the same data, if it seems different to the referee then this is indeed due to the different scale. Regarding residual oscillations in the red data, the spoiler field is actually very effective at eliminating the oscillations. As it is tricky to judge whether there are remnant oscillations in the elastic measurement from the field domain data, a comparison of the Fourier transforms of the two signals was added in the supplementary information section (Fig. 1 in the SI). Comparing the two spectra shows that the remaining intensity of the elastic signal oscillations at the relevant frequencies is actually either very close or within the noise level. We have added the following text in the manuscript to direct readers to this on page 5:

Fourier transforms of the two signals are presented in Fig. 2 of the Supplementary Information, further demonstrate that the spoiler results in negligible oscillation intensities of the elastically scattered signal, at the relevant frequencies.

Fig 3d shows the alignment. Why the alignment is zero at the critical field of 0.9 gauss meter is not addressed?

We are not sure what the referee means by a “critical field” of 0.9 gauss metre in figure 3d. Perhaps they are asking why a maxima in the signal (for example the one at 0.9 gauss meter) , which represents a maximum in the de-excitation probability, can take place in a field region where the alignment is approximately 0? If this is the question then the answer is that the excitation probability doesn't depend just on the populations of the impinging molecules and definitely not just on their alignment. Changing B_1 , changes both the amplitudes and the phases of the superposition state describing the molecules which reach the surface. The scattered state, related to the incoming state through the S-matrix, changes as well and so does the probability of scattering into a specific rotationally inelastic diffraction channel. This point is explained in the paragraph starting at the end of page 6 and continuing into page 7.

In the bottom paragraph of page 6 reference is made to two oscillation patterns. Are they on both sides of $B_1=0$? In that case the difference is hardly apparent and should be shown superimposed since it appears to be an important observation.

The oscillation patterns we were referring to are that of the calculated alignment and that of the measured $J = 2 \rightarrow J = 0$ signal. The difference between these two patterns, especially around the $B_1=0$ region are easily visible to the eye so there is no need to superimpose them. However, we have changed the text to clarify what is being compared as follows on page 6.

However, there are clear discrepancies between the oscillation of the alignment (Fig. 3D) and the oscillation of the experimentally measured signal related to the rotational de-excitation probability (Fig. 2B), discrepancies which are most noticeable at low values of $|B_1|$.

The referee was also not able to understand the meaning of Table 1 and why there are large differences in the columns labeled DFT values and Fit values. Is it because as they report on the middle of page 8 “ Consequently, we should treat the best fit S-matrix values in Table 1 as an example which could explain the data.....”?

The differences between the DFT based S matrix values and the best fit values illustrate the fact that the DFT based estimations of the S matrix elements cannot reproduce the oscillation pattern measured in the experiment (Fig. 4A). This means that any set of S-matrix values which is capable of simulating the signal is expected to be considerably different. The best fit values appearing in table 1, illustrate this point by fitting the measurements quite nicely (see Fig. 4B). The sentence the referee is citing stresses the point that the best fit S-matrix elements are not a unique set of parameters, there is more than one combination of amplitudes and phases which

can be used to fit the data. Hence, the experimental data serves as a way of testing theoretical predictions for the S-matrix rather than a way of uniquely determining them.

We envisage that the changes in the text on page 10, which were made in response to a previous comment the referee made, will help make this point clearer:

As the only unknown when simulating the experiments are the S-matrix elements, the accuracy of theoretical models can be tested by their ability to fully reproduce the oscillation patterns of the experimental signal.

In trying to better understand the article the referee looked up another Nature Comm, Ref 14 by the same group which he found well written and fully understandable and helped the referee to better understand the article under review. In summary, the results are interesting but the present manuscript is not suited for a letters journal article. With more detail the results surely would be publishable in a journal devoted to longer articles.

We agree that the experimental technique is unique, complex and not easy to understand. This obviously makes the paper harder to understand and harder to review. However, it is this complexity and uniqueness which also makes the technique capable of performing experiments which were previously impossible, producing results which in our view are interesting, surprising and important enough to be published in a journal which has a broad readership such as Nature Communications.

REVIEWERS' COMMENTS

Reviewer #1 (Remarks to the Author):

I am very pleased with the author's response to my concerns and have only one additional suggestion regarding my point #3. The authors now include nice discussion on the possible effects of the lattice distortion and electronic friction and cite several relevant theoretical papers. I would like to suggest referencing the theoretical work of R. J. Maurer, Y. Zhang, H. Guo, and B. Jiang, Faraday Disc. 214, 105 (2019)., in which the influence of electronic friction on mJ-specific inelastic scattering probabilities of H₂ from Ag(111) are discussed.

Reviewer #2 (Remarks to the Author):

I am gratified that the authors addressed seriously the objection I raised, and I am fully satisfied with the present text.

The comments from the Reviewers are reproduced in black below, with our replies written in red. We thank the Reviewers for their comments, and address each of them in turn below.

Reviewer #1 (Remarks to the Author):

I am very pleased with the author's response to my concerns and have only one additional suggestion regarding my point #3. The authors now include nice discussion on the possible effects of the lattice distortion and electronic friction and cite several relevant theoretical papers. I would like to suggest referencing the theoretical work of R. J. Maurer, Y. Zhang, H. Guo, and B. Jiang, Faraday Disc. 214, 105 (2019)., in which the influence of electronic friction on mJ-specific inelastic scattering probabilities of H₂ from Ag(111) are discussed.

We thank the reviewer for their support and have added the suggested reference (number 38) on page 10 of the revised manuscript.

Reviewer #2 (Remarks to the Author):

I am gratified that the authors addressed seriously the objection I raised, and I am fully satisfied with the present text.

We thank the reviewer for their acknowledgement that appropriate changes have been made.